# Experimental Chemotherapy-Induced Mucositis: A Scoping Review Guiding the Design of Suitable Preclinical Models

**DOI:** 10.3390/ijms232315434

**Published:** 2022-12-06

**Authors:** Junhua Huang, Alan Yaw Min Hwang, Yuting Jia, Brian Kim, Melania Iskandar, Ali Ibrahim Mohammed, Nicola Cirillo

**Affiliations:** Melbourne Dental School, The University of Melbourne, Carlton, VIC 3053, Australia

**Keywords:** chemotherapy, mucositis, oral mucositis, gastrointestinal mucositis, animal models, 5-fluorouracil, irinotecan, platinum-based drugs, methotrexate, intravenous, intraperitoneal

## Abstract

Mucositis is a common and most debilitating complication associated with the cytotoxicity of chemotherapy. The condition affects the entire alimentary canal from the mouth to the anus and has a significant clinical and economic impact. Although oral and intestinal mucositis can occur concurrently in the same individual, these conditions are often studied independently using organ-specific models that do not mimic human disease. Hence, the purpose of this scoping review was to provide a comprehensive yet systematic overview of the animal models that are utilised in the study of chemotherapy-induced mucositis. A search of PubMed/MEDLINE and Scopus databases was conducted to identify all relevant studies. Multiple phases of filtering were conducted, including deduplication, title/abstract screening, full-text screening, and data extraction. Studies were reported according to the updated Preferred Reporting Items for Systematic reviews and Meta-Analyses Extension for Scoping Reviews (PRISMA-ScR) guidelines. An inter-rater reliability test was conducted using Cohen’s Kappa score. After title, abstract, and full-text screening, 251 articles met the inclusion criteria. Seven articles investigated both chemotherapy-induced intestinal and oral mucositis, 198 articles investigated chemotherapy-induced intestinal mucositis, and 46 studies investigated chemotherapy-induced oral mucositis. Among a total of 205 articles on chemotherapy-induced intestinal mucositis, 103 utilised 5-fluorouracil, 34 irinotecan, 16 platinum-based drugs, 33 methotrexate, and 32 other chemotherapeutic agents. Thirteen articles reported the use of a combination of 5-fluorouracil, irinotecan, platinum-based drugs, or methotrexate to induce intestinal mucositis. Among a total of 53 articles on chemotherapy-induced oral mucositis, 50 utilised 5-fluorouracil, 2 irinotecan, 2 methotrexate, 1 topotecan and 1 with other chemotherapeutic drugs. Three articles used a combination of these drugs to induce oral mucositis. Various animal models such as mice, rats, hamsters, piglets, rabbits, and zebrafish were used. The chemotherapeutic agents were introduced at various dosages via three routes of administration. Animals were mainly mice and rats. Unlike intestinal mucositis, most oral mucositis models combined mechanical or chemical irritation with chemotherapy. In conclusion, this extensive assessment of the literature revealed that there was a large variation among studies that reproduce oral and intestinal mucositis in animals. To assist with the design of a suitable preclinical model of chemotherapy-induced alimentary tract mucositis, animal types, routes of administration, dosages, and types of drugs were reported in this study. Further research is required to define an optimal protocol that improves the translatability of findings to humans.

## 1. Introduction

Alimentary tract mucositis is a major adverse effect of patients receiving cancer chemotherapy treatments [1]. It is collectively referred to as chemotherapy-induced mucosal injury that affects the entire alimentary canal from the mouth to the anus, where oral mucositis (OM) and intestinal mucositis (IM) refer to painful inflammation and ulceration of the oral cavity and lower GI tract, respectively. Symptoms of this debilitating condition include ulceration, oral and abdominal pain, and can lead to a delay or even cessation of cancer treatment. Ancillary associated complications include anorexia, vomiting, and diarrhoea that remain a significant burden for 40–80% of patients undergoing chemotherapy treatment. Such obstructions to food and water intake may lead to weight loss [2]. 

There is currently no effective single intervention that can prevent or treat mucositis [3]. Being able to alleviate the symptoms of mucositis is important for several reasons. Firstly, this condition affects approximately 80% of cancer patients receiving chemotherapy and radiotherapy. Secondly, pain and decreased oral function often persist for a long period of time after therapy has concluded. Thirdly, mucositis increases mortality and morbidity which can contribute to rising healthcare costs [1]. However, despite mucositis’s significant clinical and economic impact, agents used in its management (anti-inflammatories, biologic response modifiers, cytoprotectants antimicrobials, antifungal) are generally only palliative, with few agents (palifermin and benzydamine) approved to date.

The exact pathogenic mechanisms of chemotherapy-induced alimentary tract mucositis are unknown, and this highlights that preclinical models are critical for the development of novel mechanism-based treatments. Mucositis involves multiple signalling pathways and processes that result in mucosal and luminal modifications to the intestine [4]. As such, many avenues have been explored to understand the multifactorial nature of mucositis but have not resulted in the development of effective mechanism-based management strategies. 

We believe that the development of novel interventions for the benefit of patients with chemotherapy-associated mucosal injury are limited partly due to the use of inconsistent preclinical models. Furthermore, although alimentary tract mucositis occurs concurrently in the same individual, oral and intestinal mucositis are often studied separately using different, organ-specific models. Hence, this scoping review aims to synthesise the currently available evidence of mucositis models and interpret the data generated. In particular, we aim to assess the published literature describing animal models of mucositis to document suitable preclinical models that can be used to assess the efficacy of novel interventions for the prevention and/or treatment of chemotherapy-induced mucosal injury of the alimentary tract in vivo.

## 2. Results

### 2.1. Search Results

The search of Scopus and PubMed databases produced 478 results in total. Twelve duplicates were removed via Endnote. Automation tools were used in Scopus and PubMed to exclude non-English, nonhuman, and review articles and 323 articles remained. These publications’ title and abstract were screened in accordance with the eligibility criteria, and 276 articles remained and were sought for retrieval. A total of 9 articles were not retrievable, and 267 articles remained. After reading the full text of the 267 articles, 16 articles were further excluded. The remaining 251 articles were assessed and included in the qualitative synthesis. The study selection process is summarised in Figure 1.

Due to the large number of included articles, the data extraction sheet is provided in the Appendix A. Details include: reference number, authors and year, title, population, intervention, control/comparator, and outcomes (PICO), biomarker, methodology and sample type, assessment of mucosal damage, details of treatment or intervention, and major findings. 

For clarity, the results reported below were stratified on the basis of the alimentary tract segment involved (oral or intestinal) and drug used.

### 2.2. Chemotherapy-Induced Intestinal Mucositis

#### 2.2.1. Assessment of Intestinal Mucositis

Metabolic data including body weight and food/water intake and disease activity parameters including diarrhoea and rectal bleeding were included in most of the studies. Diarrhoea severity was measured by stool consistency, and body weight was expressed as weight loss over the experimental period. Stool consistency was commonly assessed via a scoring system, in which score 0 was normal stool and score 3 and 4 corresponded to severe diarrhoea and watery stool. Most studies reported that diarrhoea severity increased and body weight decreased during the course of intestinal mucositis.

The most common methodology used amongst the studies for histopathological and morphometric evaluation, documented in 188/205 (91.7%) of the studies, was haematoxylin–eosin staining (H&E) and subsequent examination by light microscopy. Villus height and crypt depth were measured to assess their histological changes in most studies. Villus height was measured from the baseline to the villus tip and crypt depth was measured from the baseline to the crypt bottom. The majority of studies found a decrease in villus height and either increase or decrease in crypt depth in the chemotherapy-treated group. In addition, various histopathological score systems were used in different studies to examine the severity of intestinal mucositis. Although each score system contains different criteria, the common pattern is that zero means normal or no damage and severity increases as the number gets larger. Histologic criteria described in the literature include: the morphology of the villi (blunting, atrophy), crypt architecture, vacuolization, crypt necrosis, an infiltration of inflammatory cells, a disruption of brush border and surface enterocytes, a dilation of lymphatics and capillaries, oedema, intestinal epithelium architecture, crypt damage (surviving crypts per millimetre and surviving cells per crypt), the presence of haemorrhagic areas, goblet cell numbers, crypt abscess formation, enterocytes mitotic figures, a thickening of the submucosal and muscularis externa layers, and the intensity of inflammation.

A macroscopic examination of the intestinal tissues was also employed in several studies and different score systems were used. The components described in the literature consist of inflammatory aspects such as erythema, haemorrhagic areas, epithelial ulcerations and abscesses, hyperaemia, and the size of the ulcerative area. Small intestine length, colon length, and mucosal thickness were also included in several studies.

Additionally, various biomarkers were investigated in the literature to help determine the severity of intestinal mucositis, with the most common being inflammatory biomarkers. The inflammatory markers investigated in the literature include IL-1β, IL-4, IL-5, IL-6, IL-10, IL-13, tumour necrosis factor (TNF-α), interferon-γ (IFN-γ), monocyte chemoattractant protein-1 (MCP-1), NF-κB, TGF-β, myeloperoxidase (MPO), CXCL1, CXCL3, and CXCL9. The most common methods to assess the change in inflammatory cytokines level such as TNF-α and IL-1β and anti-inflammatory cytokines such as IL-4 and IL-10 are real-time PCR and enzyme-linked immunosorbent assay (ELISA). Most studies found an increased level of inflammatory cytokines and a reduction in anti-inflammatory cytokines after chemotherapeutic treatment. Interestingly, one study reported an increase of IL-4 and IL-10, at 31.5% and 39.6%, respectively [5]. MPO activity was measured by a predetermined method in other studies. Briefly, the most common protocol used was adding a solution of o-dianisidine dihydrochloride or o-dianisidine hydrochloride and hydrogen peroxide into the intestinal tissue supernatant. The MPO activity was measured by the change in absorbance at 450 nm. The level of MPO activity increased overall during the course of intestinal mucositis. Apoptosis markers were also used to assess the level of apoptosis in the crypt cells. The terminal deoxyribonucleotide transferase (TdT)-mediated nick-end labelling (TUNEL) assay was the most common method to quantify the apoptotic cells in the intestinal crypt, calculated by dividing the number of apoptotic cells by the total number of cells in the selected villi or crypt. The most common method used to assess the apoptotic assay was using an in situ apoptosis detection kit followed by IHC. The number of apoptotic cells during the course of intestinal mucositis was found to be increased in the majority of studies. The expression of caspase-3 and the Bax/Bcl-2 ratio were also used to determine the enterocyte survival. The most common method used to assess the level of caspase-3 was immunohistochemistry. IHC, western blotting, and RT-PCR were documented for the assessment of Bax and Bcl-2 levels. The third category of biomarkers investigated in intestinal mucositis are intestinal oxidative stress biomarkers, which include a number of antioxidants: malonaldehyde (MDA), glutathione (GSH), catalase (CAT), superoxide dismutase (SOD), and glutathione peroxidase (GPx). The methods used to measure the level of oxidative stress markers were pre-established by other studies. Most studies reported a decrease in GSH, CAT, and SOD and an increase in MDA. However, one study reported a 3.81-fold increase in the level of GSH after chemotherapy [6]. Intestinal proliferation markers were also employed in some studies to assess the effect of mucositis on epithelial intestinal cells, including proliferating cell nuclear antigen (PCNA) and Ki-67. The most common method to measure PCNA and Ki-67 was via IHC. Most studies found a negative effect of chemotherapy on PCNA and Ki-67 and the reduction of PCNA positive cells. Two studies reported a significant increase in Ki-67 positive cells after FOLFOX chemotherapeutic regimen [7,8]. Sucrase activity was used by several studies as an indicator for small intestinal damage, which was measured by a ^13^C-sucrose breath test. Biomarkers were also used to assess the level of intestinal permeability as a result of mucositis; such biomarkers include the tight junction molecules ZO-1, occludin, junctional adhesion molecule-A, and Claudin-2, as well as other biomarkers such as diamine oxidase (DAO) and endotoxin. Some studies also suggested that DAO could be used to quantify intestinal mucosal injury [9,10]. The most common method to assess the level of ZO-1 and occludin was via IHC and RT-PCR and most studies found that chemotherapy induced negative changes to ZO-1 and occludin. ELISA was most commonly used to determine the level of DAO. A reduction of plasma DAO was reported in the majority of studies. It was also found that there was a dose–response relationship between DAO and the dosage of 5-Fluorouracil [11].

#### 2.2.2. Animal Models of Intestinal Mucositis Related to Specific Chemotherapeutic Drugs

##### 5-Fluorouracil (5-FU)

A total of 103 articles documented 5-FU-induced intestinal mucositis in animal models. A summary of findings is presented in Table 1. Of the 103 articles, 72 used mouse models, 30 studies used rats, and 1 used domestic pig models. Different dosages and routes of administration of the drug were employed. Most articles used intraperitoneal (i.p.) injection with a dosage range between 25 and 450 mg/kg. In fewer studies, mucositis was induced by 5-FU with a range of 20–50 mg/kg orally by gavage or dropsonde into the oesophagus of animal models, except pig models, who were fed by oral bolus. Only three articles used intravenous (i.v.) injection or infusion with a range of 50–200 mg/kg. In general, a similar dosage range of 5-FU was found among three types of animal models, although an outlier at 100 ng/kg was found in one paper reporting a combination of 5-FU with doxorubicin [8]. Moreover, the dosage of each injection chosen in these articles was influenced by the number of injections. In particular, most studies adopted a single dosage of 5-FU of 150 mg/kg and above to induce intestinal mucositis, whereas multiple injections were made with dosages of 100 mg/kg or less. A summary of the articles is presented in Table 1.

##### Irinotecan

A total of 34 articles produced drug-induced gastrointestinal mucositis in rat and mouse models with irinotecan. Among the 34 articles, 22 articles used mouse samples whilst the remaining 12 articles used rat samples. The concentration of irinotecan used varied among these studies, from a minimum of 10 mg/kg to a maximum of 270 mg/kg via i.p. on mouse models, and a minimum of 20 mg/kg to 200 mg/kg via intraperitoneal injection on rat models. The use of irinotecan was found to decrease the body weight and food intake of the animals in these experimental models. Diarrhoea was often seen as a result of dose-limiting side effects. A summary of the articles is shown in Table 2.

##### Platinum-Based Chemotherapy Drugs

Platinum-based drugs that were used included cisplatin, carboplatin, and oxaliplatin. A total of 251 in vivo articles were reviewed, and 16 articles reporting drug-induced gastrointestinal mucositis on rat and mouse models with platinum-based chemotherapy drugs were included. A summary of the articles is shown in Table 3. Among the 16 articles, 10 articles used mouse models whilst the remaining 6 articles used rats. For the mouse models, only one article used carboplatin (100 mg/kg intraperitoneal injection), five articles used cisplatin (2 mg/kg i.p. to 11 mg/kg i.p.), and four articles used oxaliplatin (1 mg/kg i.p. to 5 mg/kg i.p.). For the rat models, the five articles used cisplatin, from a minimum of 5 mg/kg i.p. to 7 mg/kg intraperitoneally. Aside from mucositis, other side effects such as diarrhoea, vomiting, and reduced body weight were often observed. A summary of the articles is shown in Table 3.

##### Methotrexate

Among the 251 included articles, 33 articles documented animal models of methotrexate-induced intestinal mucositis. Within the 33 articles, 6 articles reported the use of mouse models, and 27 articles reported the use of rat models. A summary of the articles is shown in Table 4.

In the six articles reporting murine models, four articles reported intraperitoneal (i.p.) injection of methotrexate (MTX) with a dosage ranging from 20 mg/kg to 500 mg/kg. The remaining two articles used subcutaneous (s.c.) injection with a dosage of 12.5 mg/kg.

In the 27 articles documenting a rat model, nine articles reported i.p. injection of MTX with a concentration ranging from 2.5 mg/kg to 90 mg/kg. Six articles used s.c. injection with a dosage ranging from 1.5 mg/kg to 3.5 mg/kg. Five articles documented the use of intramuscular (i.m.) injection of MTX with a dosage of 1.5 mg/kg. Five articles reported intravenous (i.v.) injection of MTX at a dosage from 20 mg/kg to 150 mg/kg. The remaining two articles reported oral intake of MTX at a dosage of 5 mg/kg.

##### Other Chemotherapeutic Agents

A total of 32 articles reported intestinal mucositis induced by other chemotherapeutic agents, including doxorubicin, capecitabine, afatinib, SN38, ailanthone, melphalan, busulfan, cyclophosphamide, paclitaxel, cytarabine (Ara-C), etoposide, ifosfamide, epirubicin, dioscrin, and S-1, a combination of 1-(2-tetrahydrofuryl)-5-fluorouracil (FT), 5-chloro-2,4 dihydroxypyridine (CDHP), and potassium oxonate (Oxo). A summary of the results is presented in Appendix A.

Doxorubicin: a mouse model was used in 10 publications [12,184,185,186,187,188,189,190,191,192], with a dosage ranging from 10 to 20 mg/kg intraperitoneally. Two articles documented the use of a rat model with a dosage of 20 mg/kg i.p. [132,193] and one article used a piglet model (100 mg/m^2^ surface area i.v.) [194]. 

Capecitabine: two articles reported a rat and murine model, respectively, both with a dosage of 500 mg/kg orally [195,196]. 

Afatinib and SN38: one article reported the use of both afatinib and SN38 in a zebrafish model, with a dosage ranging from 10 to 40 µg/g orally and 10 to 40 µg/g intraperitoneally, respectively [197]. 

Ailanthone: one article documented the use of ailanthone in a mouse model with a dosage of 2–39.8 mg/kg orally [198].

Melphalan: one publication reported a mouse model with a concentration from 85.5 to 95.7 mg/m^2^ i.p. [199].

Busulfan: a piglet model was used in one article with a dosage at 12.8 mg/kg i.v. [200]. A mouse model was documented in one article with a dosage at 40 mg/kg, but the route of administration was not specified [201].

Cyclophosphamide: five articles reported a mouse model with a dosage ranging from 50 to 550 mg/kg i.p. [127,202,203,204,205]. One article reported a rat model with a dosage at 120 mg/kg i.p. [132] and one article reported a piglet model with a dosage at 120 mg/kg i.v. [200].

Paclitaxel: mouse models were employed in one article, with a dosage ranging from 2 to 4 mg/kg i.v. [206].

Cytarabine (Ara-C): one article used a mouse model with a drug concentration at 3.6 mg/mouse i.p. [207] and one article reported a rat model with a dosage at 30 mg/kg s.c. [9].

Etoposide: a rat model was used in one publication with a dosage at 40 mg/kg i.p. [132].

Ifosfamide: one article reported the use of a rabbit model with a dosage ranging from 30 to 60 mg/kg i.v. [208].

Epirubicin: one study reported the use of epirubicin as a chemotherapeutic agent in a mouse model at a dosage of 12 mg/kg i.p. [209].

Dioscrin: dioscrin at a dosage of 60 mg/kg administered intragastrically was documented in one study using a rat model [148].

S-1: S-1, an oral fluorouracil comprising 1-(2-tetrahydrofuryl)-5-fluorouracil (FT), 5-chloro-2,4-dihydroxypyridine (CDHP), and potassium oxonate (Oxo) in a molecular ratio of 1:0.4:1, was reported to be used in a rat model at a dosage of 20 mg/kg orally [10].

### 2.3. Chemotherapy-Induced Oral Mucositis

#### 2.3.1. Assessment of Oral Mucositis

The most common method used to evaluate histopathological features of oral mucositis was the H&E staining of the oral mucosa, tongue tissues, or cheek pouch, documented in 35/52 articles (67.3%). In the literature, various histological score systems were used to classify the severity of mucositis, -including the epithelial architecture, membrane integrity, intensity of inflammatory cell infiltration, connective tissue organisation, oedema of the submucosa, intensity of inflammation, vasodilation, haemorrhage, oedema, ulcers, and abscesses.

In addition, a macroscopic examination was used in a number of studies to assess mucosal injury. The macroscopic features used in the literature were similar to the criteria employed in the assessment of intestinal mucositis, which included inflammatory aspects such as erythema, hyperaemia, haemorrhagic areas, epithelial ulcerations, and abscesses, hyperaemia, and the dimension/area of the ulcerative area. Oral epithelial thickness and cheek pouch thickness were also used. Overall, a varying degree of reduction in epithelial thickness, architecture, and structure were detected. An increase in the abundance of inflammatory infiltrates was commonly reported.

The use of biomarkers was similar to the studies on intestinal mucositis. The categories of biomarkers documented included inflammatory cytokines (MPO, TGF-β, IL-1β, NF-κB, TNF-α, IL-1, IL-6), apoptosis markers (Bcl-2, caspase-3), proliferation markers (Ki-67, PCNA), and oxidative stress markers (MDA, GSH, SOD, CAT). The most common method to assess the change in inflammatory cytokines level such as TNF-α and IL-1β and anti-inflammatory cytokines such as IL-6 and IL-10 is an enzyme-linked immunosorbent assay (ELISA). Most studies found an increased level of inflammatory cytokines and a reduction in anti-inflammatory cytokines after chemotherapeutic treatment. MPO activity was measured with a similar protocol to MPO assays addressed in intestinal mucositis. The level of MPO activity increased overall during the course of oral mucositis. Metabolic data such as a reduction in body weight were also included in most of the studies. 

#### 2.3.2. Animal Models of Oral Mucositis Related to Specific Chemotherapeutic Drugs 

##### 5-Fluorouracil

A total of 50 studies documented oral mucositis induced by 5-FU in animal models. A summary of findings is presented in Table 5. Of the 50 articles, 13 used mice, 8 studies used rats and 29 used hamsters. Different dosages and routes of administration of the drug were employed, most articles reported using an intraperitoneal (i.p.) injection with a dosage range between 10 and 150 mg/kg body weight. One article reported i.v. administration of 5-fluorouracil [44], and one study did not specify the route of administration [210]. The total dosage of injections was dependent on the number of injections given to the model. Most studies adopted a double dosage of 5-FU at 40 mg/kg and then 60 mg/kg in subsequent days.

##### Irinotecan

A total of two articles reviewed utilised a drug-induced model of oral mucositis using irinotecan. Both these studies were on rats and the concentration of the drug used was 200 mg/kg body weight at a single dose. In both studies, irinotecan was administered intraperitoneally. One study utilised a multiple intervention model using irinotecan, methotrexate, and 5-FU [108]. The other study used irinotecan only as the intervention [254]. 

##### Topotecan

Of the oral mucositis models, only one used topotecan to induce mucositis. This article utilised rabbits as the animal model. The concentration of the drug used was 0.5 mg/kg of body weight and was administered via intravenous bolus [255]. 

##### Methotrexate

A total of two articles utilised methotrexate to induce a model of oral mucositis. Both of these studies were on rats and the concentration of the drug utilised was 1.5 mg/kg of body weight, administered intramuscularly [88,108]. 

##### Other Chemotherapeutic Agents

One article reported the use of a combination of drugs including busulfan and cyclophosphamide on pig models. Busulfan and cyclophosphamide were administered intravenously with a total dosage of 12.8 mg/kg and 120 mg/kg, respectively [200].

#### 2.3.3. Induction of Oral Mucositis in Animal Models

Mechanical-, chemical-, or radiation-induced injury of the oral mucosal surface is additionally required in most existing chemotherapy-induced oral mucositis animal models when chemotherapy is intraperitonially administered. Of the 53 articles reporting oral mucositis, 33 articles introduced mechanical irritation to the cheek pouch such as superficial scratching with the tip of an 18-gauge needle. Eight articles reported the use of a chemical agent such as acetic acid for ulcer induction in animal models. Two articles report i.v. administration of 5-FU to induce oral mucositis in mice without any additional stimuli [44,142]. Another article also reported i.v. administration of topotecan to induce oral mucositis in rabbits without any additional stimuli [255]. Nine articles did not report any form of additional stimulating treatment other than the i.p. chemotherapeutic agents. However, when i.p. administration of 5-FU was used without additional intervention to induce oral mucositis, no visible ulcerations were recorded or observed in the oral cavity. A summary of the findings is presented in Table 6. 

## 3. Discussion

Our scoping review includes data from 251 publications involving chemotherapy-induced intestinal and oral mucositis in animal models. Importantly, the vast majority of these models did not produce intestinal and oral mucositis together. Only seven articles [44,70,88,99,108,109,200] were chemotherapy-induced alimentary tract mucositis. 

For chemotherapy-induced intestinal mucositis models, the review found that the most commonly used animals were mice, while the most commonly used drug was 5-FU. The most common route of administration was intraperitoneal (i.p.) injection (25–450 mg/kg).While this method is beneficial because it allows the absorption of large amounts of the intervention rapidly, the disadvantage is that the drug can have a large variability in effectiveness and misinjection [256]. Another problem associated with the i.p. mouse model is that the intestinal mucosa is more sensitive to chemotherapy compared to the upper alimentary tract such as the oral cavity. As a result, the intestinal function deteriorates quickly, and mice need to be euthanized. This usually occurs before any oral lesions develop, which makes it difficult to study oral mucositis [257]. For oral mucositis models, the most common animal models used were hamsters with i.p. injection of 5-FU (40–100 mg/kg). Most models (75%) applied additional mechanical or chemical irritation with chemotherapeutic drugs to induce oral mucositis. Hence, the majority of animal models did not reproduce a common clinical scenario in that they did not use the same route of administration of the antineoplastic agent as in patients and/or applied additional, nonphysiological stimuli. 

### 3.1. Translatability of Models

Preclinical models evaluated in the literature mainly induced intestinal mucositis. There was no standardisation to the dosages administered, and the literature also reported that the dosage of each injection depended on the number of injections. In humans, the dosage for chemotherapy treatments is dependent on individual tolerances to the treatment. Thus, most dosages are calculated based on human surface area, while the dosage calculation for preclinical models is dependent on weight. As such, for the preclinical data to be translatable, a standardised conversion is required. For example, in the FOLFIRI regimen for colorectal cancer, the standard dosage of 5-FU is 2400 mg/m^2^, and it was calculated that a dosage of 400 mg/kg in rats approximated 2222 mg/m^2^ in humans [103]. However, the 400 mg/kg dosage sits in the high end of the dosage range reported in the literature, and most studies had doses lower than 400 mg/kg. Nevertheless, the equivalent dosage of human doses to other animal models still needs more research. Therefore, whether a lower dosage of chemotherapeutic drugs used in animal models can mimic human disease is questionable. 

While 52 articles demonstrated the establishment of chemotherapy-induced oral mucositis, mechanical or chemical irritation was commonly used in the literature to induce oral mucositis in these animal models, in addition to the i.p. administration of chemotherapeutic agents. Only one study [44], where the chemotherapeutic agents were administered intravenously, produced ulceration lesions in the oral cavity and replicated oral and intestinal mucositis both macroscopically and histologically, without using any additional stimuli. It was suggested that intraperitoneal 5-FU injection in mouse models can induce intestinal mucositis reproducibly but did not affect the oral mucosa significantly, hence local mechanical trauma or chemical injury were included in animal oral mucositis models [257]. This may be likely attributed to the highly keratinized nature of oral epithelium in mice, thus rendering it less susceptible to mucosa breakage [258]. Furthermore, i.p. administration increases the concentration of chemotherapy in the peritoneal cavity and targets primarily organs in the peritoneal cavity, thus toxicity in these organs exceeds that at other sites [257]. It was supported by Yamaguchi et al. that chemotherapeutic drugs did not induce oral mucositis directly and an additional mechanical or chemical injury was needed [99]. However, with the addition of a mechanical or chemical injury, whether the direct effects of chemotherapeutic drugs on oral mucosa can be examined is questionable. The standardisation of such mechanical and chemical injury protocols is difficult [257]. Whether the additional mucosa manipulation mimics the course of oral mucositis in humans is also questionable. Bertolini et al., studied the possibility of inducing oral mucositis without additional stimuli [257]. It was found that a high daily dose of 5-FU i.p. was unable to induce oral mucositis, but the mouse model with 50 mg/kg 5-FU i.v. injected every 48 h for 13 days showed macroscopical and histological features and an inflammatory cytokine profile consistent with oral mucositis [257]. 

### 3.2. Pathophysiology

The pathophysiology of intestinal mucositis might involve a combination of villi length and crypt depth changes, oxidative stress, apoptosis, inflammatory reactions, a proliferative capacity of the intestinal cells, and the composition of the gut microbiome [8]. Sonis et al. proposed a five-stage model of the pathogenesis of mucositis induced by chemotherapeutic agents, which included initiation, upregulation and message generation, signalling and amplification, ulceration and inflammation, and the healing stage [259].

The production of reactive oxygen species and the increase in oxidative stress were suggested to contribute to the initiation phase, subsequently leading to cell death and DNA damage [260]. Rtibi et al., found that the antioxidant biomarkers SOD, CAT, and GPx decreased in 5-FU and CAP-induced intestinal mucositis in a rat model, confirming the role of oxidative stress in the onset of intestinal mucositis [195]. Consistently, the use of antioxidants has been found to reduce the severity of mucosal injury in vitro, in vivo, and in clinical studies [261]. 

The second phase is characterised by the involvement of inflammatory components, with NF-kB being the key molecule. The activation of proinflammatory cytokines can lead to tissue injury and cell apoptosis in both the intestinal crypt and oral mucosa [259]. Multiple studies examined the role of inflammatory cytokines in the pathophysiology of mucositis. Logan et al., reported an increase in NF-κB, TNF, IL-1β, and IL-6 serum concentration following administration of 5-FU, MTX, and irinotecan in an alimentary tract rat model, but the serum concentration changes followed histological changes in most instances. Both oral and intestinal histological changes were investigated in these studies [108]. It was also found that inflammasome activation following oxidative stress contributed to the pathogenesis of an irinotecan-induced intestinal mucositis mice model. Inflammasome activation was crucial to activate IL-1β and IL-18 [118].

The third stage is characterised by the amplification of inflammatory response by proinflammatory cytokines such as TNF-α [259]. In a 5-FU-induced oral mucositis hamster model, TNF-α inhibitors were found to reduce the severity of oral mucositis. TNF-α can modulate the expression of other cytokines such as IL-1β and IL-6 and affect the apoptosis and survival of other cells, hence it was postulated that TNF-α was important in the pathogenesis of oral mucositis [228].

The fourth ulcerative phase is considered to be the culmination of the mucositis process. The apoptosis of epithelial stem cells and tissue injury as a result of inflammation and oxidative stress result in the breakdown of mucosa and the loss of epithelial integrity observed in oral and intestinal mucositis. The breakdown of mucosa not only allows bacteria translocation, but the bacterial products can further enhance the inflammatory response by activating macrophages in the local environment.

The last healing phase is characterised by a restoration of the proliferation capacity and local microbiome [259]. In a 5-FU-induced intestinal mucositis rat model, bacterial translocation to the mesenteric lymph nodes was significant after 5-FU administration and was speculated to be the result of inflammatory cytokines release and mucosal injury [93]. Al-Azri et al. reported the involvement of matrix metalloproteinases MMP-3 in the initiation of inflammatory reactions and it may be associated with the production of NF-κB and inflammatory cytokines [254]. MMP-9 might be involved in the last healing phase in the 5-phase model of mucositis [254]. Sonis et al., summarised that the mechanisms of oral and gastrointestinal tract mucositis were likely to be similar, but further research was still required [259].

### 3.3. Optimal Dosage

From the publications that were included in the current scoping review, there were large differences between studies in terms of the dosage used and injection regime to induce mucositis in animal models, hence the total doses injected varied widely. Such variations were largest in intraperitoneal drug administration. One of the studies included in the present review by Zhang et al., investigated the optimal dosage range of a total of five dosages of 5-FU to induce intestinal mucositis in mouse models [11]. The study proposed the criteria of the optimal animal model as follows: it should closely mimic the mucosal injury in humans, and it should have an acceptable survival rate in order to perform any treatment and investigations on these animal models. The study suggested that five dosages of 50 mg/kg to 100 mg/kg i.p. per day fulfilled the above criteria, and a dosage lower or higher than this concentration could either result in insufficient mucosal injury or high mortality rate [11]. Another study conducted by Fijlstra et al., investigated the optimal dosage of a single dose of MTX in a rat model in a pilot experiment. The criteria were similar to Zhang et al., which included pronounced mucosal injury and low mortality rate. The experiment found that the severity of mucositis increased with increasing dosage, but the mortality rate increased at the same time [11]. They proposed a single dosage of 60 mg/kg of MTX i.v. was the optimal dosage to induce intestinal mucositis in a rat model [174]. The standardised optimal dosage of different chemotherapeutic drugs and the route of administration in different animal models require further research.

### 3.4. Limitations

Our study attempted to gain a clear understanding of the advancements made in animal models of mucositis, with the eventual aim to translate such results in humans. Our review assessed the current understanding of interventions against mucositis-induced preclinical models. However, our exclusion criteria, such as excluding non-English articles or search string (excluding radiotherapy or radiation in the title or abstract) could mean that we missed out on potential information that could provide us the complete bigger picture of this condition. 

Within the articles themselves, there were some limitations. For example, many articles did not have enough statistical power as the sample size was generally low. Some studies reported *n* < 5 per experimental group. Furthermore, the risk of selection bias could not be ruled out among animal models. As the baseline characteristics such as the species, age, weight, and gender were not accounted for when determining drug concentrations. Future studies can iterate on these limitations by exploring the optimal dosage required, such as in terms of single or multiple doses over a period of time to mimic mucosal injury in humans while maintaining a high survival rate.

## 4. Methods

### 4.1. Protocol and Search Strategy

The results of the scoping review are reported in accordance with the updated Preferred Reporting Items for Systematic reviews and Meta-Analyses (PRISMA) guidelines [5]. The following search strategy was used to search for related articles in Scopus and PubMed databases and was conducted in May 2022:

(((mucositis[Title/Abstract] OR mucosal injury[Title/Abstract]) AND (oral[Title/Abstract] OR intestinal[Title/Abstract] OR alimentary[Title/Abstract])) AND (chemotherap*[Title/Abstract] OR cancer treatment[Title/Abstract] OR cancer therapy[Title/Abstract] OR antineoplas*[Title/Abstract])) AND (mouse[Title/Abstract] OR mice[Title/Abstract] OR rat[Title/Abstract] OR hamster*[Title/Abstract] OR preclinical[Title/Abstract] OR pre-clinical[Title/Abstract] OR animal*[Title/Abstract]) NOT radiotherapy NOT radiation.

### 4.2. Eligibility Criteria

Articles were included in the review if they met the following criteria: (a) any peer-reviewed article presenting data such as original article, short communication, and research letter, (b) articles written in English; and (c) research using animals that were administered traditional chemotherapeutic agents. Articles with the following criteria were excluded: (a) articles not reporting original data such as reviews and systematic reviews; (b) articles not documenting animal models of chemotherapy-induced mucositis; (c) in vitro or human studies; (d) articles documenting immunotherapy-induced mucositis; and (e) articles not assessing mucosal damage in animal models.

### 4.3. Data Selection and Collection

Before title/abstract screening, automation tools in Scopus and PubMed were used to exclude review articles, nonhuman studies, and non-English articles. The articles were then deduplicated. A total of 5 reviewers were involved in the screening process. Each of the remaining articles were screened by 2 independent reviewers by reading the title and abstract. Articles were included or excluded according to the eligibility criteria. Cohen’s Kappa score was calculated to be 95.4, which showed a strong agreement between assessor pairs. Any discrepancy in the title/abstract screening between the two reviewers was resolved after discussion with the research supervisor (N. C.). A full-text screening for the included articles after title/abstract screening was then performed. Each article was screened and assessed against the eligibility criteria. Data extraction was performed and tabulated for the included articles. The extracted parameters were author, year, title, population studied, intervention used to induce mucositis, control, outcome and measurement, biomarkers, methodology and sample type, assessment of mucosal damage, details of treatment or intervention, and major findings.

## 5. Conclusions

The main purpose of this scoping review was to document and evaluate evidence from the entirety of the literature on chemotherapy-induced mucositis animal models. These in vivo models are used to elucidate the pathophysiology of mucositis and to develop potential interventions to alleviate or prevent mucositis. Our analysis showed that there were consistent data that may inform the development of a more standardised animal model that can help identify the pathophysiology of chemotherapy-induced mucositis. These models may further be used to detect, prevent, or ameliorate the mucosal toxicity of antineoplastic treatments. To help design a chemotherapy-induced alimentary tract mucositis animal model, all the studies reported here included animal types, routes of administration, dosages, and types of drugs. The results suggest that the use of a rodent model of chemotherapy reproducing the clinical and histologic features of both oral and intestinal mucositis without additional noxious stimuli (e.g., repeated cycles of 50 mg/kg 5-FU intravenously) may represent a useful in vivo preclinical model for studying chemotherapy-induced alimentary tract mucositis. However, there is still a large variation among studies and further research is required to define optimal experimental conditions for a reproducible animal model that is suitable for preclinical studies. 

## Figures and Tables

**Figure 1 ijms-23-15434-f001:**
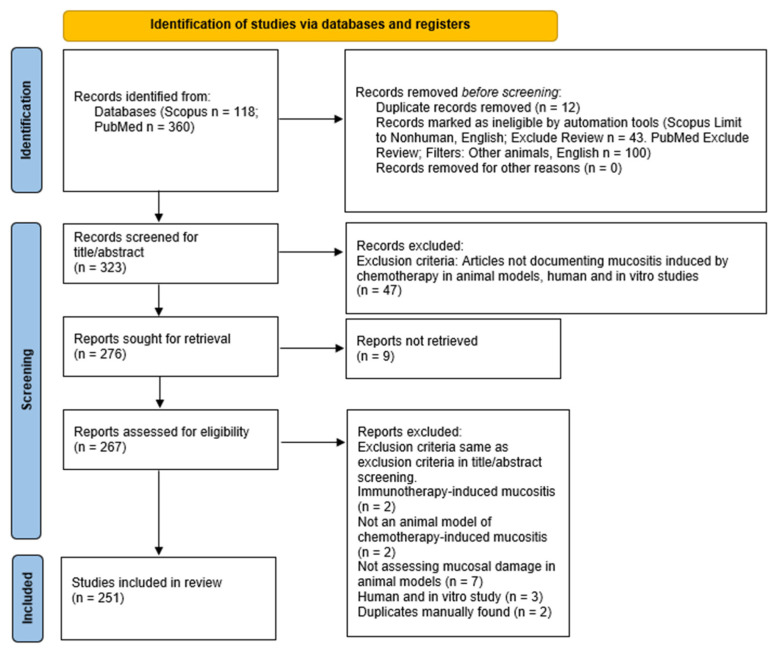
Flowchart of the data collection and selection process in accordance with PRISMA-ScR guidelines.

**Table 1 ijms-23-15434-t001:** Summary of data collected from in vivo animal studies involving 5-fluorouracil-induced intestinal mucositis. Drug concentrations obtained from studies are listed in a range of minimum to maximum dosage administered in animal models.

Animal Models	Number of Articles	Dosage of Drug	References
Mice	72	100 ng/kg i.p. ^a^; 25–450 mg/kg i.p.; 50–200 mg/kg i.v. injection/infusion; 50 mg/kg orally.	[7,8,11,12,13,14,15,16,17,18,19,20,21,22,23,24,25,26,27,28,29,30,31,32,33,34,35,36,37,38,39,40,41,42,43,44,45,46,47,48,49,50,51,52,53,54,55,56,57,58,59,60,61,62,63,64,65,66,67,68,69,70,71,72,73,74,75,76,77,78,79,80]
Rats	30	40–400 mg/kg i.p.; 20–50 mg/kg orally;	[81,82,83,84,85,86,87,88,89,90,91,92,93,94,95,96,97,98,99,100,101,102,103,104,105,106,107,108,109]
Pigs	1	12 mg/kg orally	[110]
**Total**	**103**		

i.p.: intraperitoneal injection as a route of chemotherapy administration, i.v.: intravenous injection. ^a^ A dose of 100 ng/kg was found in one paper reporting a combination of 5-FU with doxorubicin.

**Table 2 ijms-23-15434-t002:** Summary of data collected from in vivo animal studies involving irinotecan (CPT-11)-induced intestinal mucositis. Drug concentrations obtained from studies are listed in a range of minimum to maximum dosage administered in animal models.

Animal Models	Number of Articles	Dosage of Drug	References
Mice	22	10–270 mg/kg i.p.	[46,111,112,113,114,115,116,117,118,119,120,121,122,123,124,125,126,127,128,129,130,131]
Rats	12	20–200 mg/kg i.p.	[108,132,133,134,135,136,137,138,139,140,141,142]
Total	34		

i.p.: intraperitoneal injection as a route of chemotherapy administration, i.v.: intravenous injection.

**Table 3 ijms-23-15434-t003:** Summary of data collected from in vivo animal studies involving platinum-based chemotherapy drugs-induced intestinal mucositis. Drug concentrations obtained from studies are listed in a range of minimum to maximum dosage administered in animal models.

	Animal Models	Number of Articles	Dosage of Drug	References
Carboplatin	Mice	1	100 mg/kg i.p.	[143]
Cisplatin	Mice	5	2–11 mg/kg i.p.	[75,144,145,146,147]
	Rats	6	5–7 mg/kg i.p.	[148,149,150,151,152,153]
Oxaliplatin	Mice	4	1–5 mg/kg i.p.	[7,8,154,155]
	**Total**	**16**		

i.p.: intraperitoneal injection as a route of chemotherapy administration, i.v.: intravenous injection.

**Table 4 ijms-23-15434-t004:** Summary of data collected from in vivo animal studies involving methotrexate-induced intestinal mucositis. Drug concentrations obtained from studies are listed in a range of minimum to maximum dosage administered in animal models.

Animal Models	Number of Articles	Dosage of Drug	References
Mice	6	20–500 mg/kg i.p.; 12.5 mg/kg s.c.	[156,157,158,159,160,161]
Rats	27	2.5–90 mg/kg i.p.; 1.5–3.5 mg/kg s.c.; 1.5 mg/kg i.m.; 20–150 mg/kg i.v.; 5 mg/kg orally	[6,86,88,108,157,162,163,164,165,166,167,168,169,170,171,172,173,174,175,176,177,178,179,180,181,182,183]
**Total**	**33**		

i.p.: intraperitoneal injection as a route of administration, i.v.: intravenous injection, s.c.: subcutaneous injection, i.m.: intramuscular injection.

**Table 5 ijms-23-15434-t005:** Summary of data collected from in vivo animal studies involving 5-fluorouracil-induced oral mucositis. Drug concentrations obtained from studies are listed in a range of minimum to maximum dosage administered in animal models, including the route of administration.

Animal Models	Number of Articles	Dosage of Drug	References
Mice	13	10–100 mg/kg i.p.; 50 mg/kg intravenously (i.v.)	[44,70,210,211,212,213,214,215,216,217,218,219,220]
Rats	8	40–150 mg/kg i.p.	[88,99,108,109,221,222,223,224]
Hamsters	29	40–100 mg/kg i.p.	[225,226,227,228,229,230,231,232,233,234,235,236,237,238,239,240,241,242,243,244,245,246,247,248,249,250,251,252,253]
Total	50		

i.p.: intraperitoneal injection as a route of chemotherapy administration.

**Table 6 ijms-23-15434-t006:** Additional stimuli used in oral mucositis animal models other than the chemotherapeutic agents.

Additional Stimuli	Number of Articles	References
Mechanical irritation	33	[70,109,217,221,222,225,226,227,228,229,230,232,233,234,235,236,237,238,239,240,241,242,243,244,245,246,247,248,249,250,251,252,253]
Chemical injury	8	[99,212,213,219,220,223,224,231]
No additional stimuli	12	[44,88,108,200,210,211,214,215,216,218,254,255]
**Total**	**53**	

## Data Availability

Full datasets are available upon reasonable request to the corresponding author.

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
