# Peer review of "Experimental Chemotherapy-Induced Mucositis: A Scoping Review Guiding the Design of Suitable Preclinical Models"

_ijms, 2022, doi:10.3390/ijms232315434_

Round 1
Reviewer 1 Report
Manuscript of considerable interest for the dental sector
Need for a minor revison
Add keywords
Introduction, add pre treatments before starting chemo therapy
Results: making them more usable for the reader, they are very confusing
Discussion: as future objectives to analyze the effectiveness of the postbiotic-based gel in reducing the incidence of lesions and having regenerating power , as already studied by the research group of
prof scribante et al
Author Response
comments attached

Reviewer 2 Report
The article deals with a scoping review of animal models of the development of chemoinduced oral and gastrointestinal mucositis. The proposal is coherent, the design and methodology of the article are appropriate, and bring significant results for new studies in animal models for chemoinduced oral mucositis.
There are some typos, and the topic is not very current, since the new treatments for cancer are immunotherapies and checkpoint inhibitors.
Author Response
comments attached

Reviewer 3 Report
This study was interesting. If this study was performed as authors' purpose, it would be an important reference in this field. However, there were several flaws to be clarified before publication. If these flaws are not corrected successfully, I can't recommend this article.
1. In the abstract, authors found 251 titles from the total of 478 hits met the inclusion criteria. Among them, 207 titles were intestinal mucositis and 52 titles oral mucositis. When summed up, they were 259 titles. It was different to the first description. Considering the authors' mention-"Importantly, the vast majority of these models did not produce intestinal and oral mucositis together", there was no mixed case.
2. In the abstract, authors stated that "To assist with the design of a suitable pre-clinical model of chemotherapy-induced alimentary tract mucositis, animal types, routes of administration, dosages, and types of drugs were reported in this study". Then, what's the best model in this study? If authors did extensive review, they should present the best model as a result. If they did not find any proper model, then the purpose of this study was not accomplished.
3. Page 2 line 72 "2hemotherapy" seemed to be typo of "chemotherapy". Please check this.
4. In method section, authors suggested some inclusion and exclusion criteria. They were selection criteria. However, authors did not suggest the criteria for the title evaluation. Without the criteria for the title evaluation, it seemed to be impossible to find the best model.
5. In results, authors introduced several assessment methods for intestinal mucositis and oral mucositis. However, these assessment methods such as mucosal thickness, markers' expression level, etc have not been evaluated by the authors. There were just introductions and listing them. In case of introducing several agents, the dosage has been introduced without mentioning their efficacy.
Author Response
comments attached

Round 2
Reviewer 3 Report
Authors addressed all issues raised by reviewer. I don't have further comment on this article.